# HIV Reservoirs and Treatment Strategies toward Curing HIV Infection

**DOI:** 10.3390/ijms25052621

**Published:** 2024-02-23

**Authors:** Kouki Matsuda, Kenji Maeda

**Affiliations:** 1Joint Research Center for Human Retrovirus Infection, Kagoshima University, Kagoshima 890-8544, Japan; 2AIDS Clinical Center, National Center for Global Health and Medicine, Tokyo 162-8655, Japan

**Keywords:** HIV, HIV reservoir, Shock and Kill, Block and Lock, HIV cure, latency-reversing agents

## Abstract

Combination antiretroviral therapy (cART) has significantly improved the prognosis of individuals living with human immunodeficiency virus (HIV). Acquired immunodeficiency syndrome has transformed from a fatal disease to a treatable chronic infection. Currently, effective and safe anti-HIV drugs are available. Although cART can reduce viral production in the body of the patient to below the detection limit, it cannot eliminate the HIV provirus integrated into the host cell genome; hence, the virus will be produced again after cART discontinuation. Therefore, research into a cure (or remission) for HIV has been widely conducted. In this review, we focus on drug development targeting cells latently infected with HIV and assess the progress including our current studies, particularly in terms of the “Shock and Kill”, and “Block and Lock” strategies.

## 1. Introduction

Nearly 40 years have passed since the discovery of acquired immunodeficiency syndrome (AIDS) caused by the human immunodeficiency virus (HIV), and advances in antiviral therapy have significantly improved the prognosis of individuals infected with HIV [1]. Antiviral therapy, specifically the combination antiretroviral therapy (cART), strongly suppresses viral replication and reduces the plasma HIV viral load, resulting in a significant reconstitution of the immune system [2,3,4]. The life expectancy of patients infected with HIV who are treated with cART has improved significantly since 1996, and mortality rates for people infected with HIV have now approached general mortality rates [5,6]. Thus, the successful development of cART has changed AIDS from an inevitably fatal disease to a manageable chronic infection. Although cART can reduce viral production in the patient’s peripheral blood to levels below the detection limit, it cannot eliminate the HIV genome integrated into the cells. Thus, even life-long cART cannot eradicate HIV because of its persistence in a latent form in cell reservoirs [7,8,9,10,11] (Figure 1). In individuals infected with HIV, viral rebound is observed within weeks of cART cessation [12,13]; hence, they require lifelong treatment. Additionally, with an increase in the number of older adults with HIV who have been taking medications for more than 20 years, various complications besides AIDS-related diseases have emerged. Under such circumstances, several studies have been conducted towards curing HIV or causing it to go into remission, and one possible approach is to use drugs that can deplete or inactivate HIV reservoir cells. In this article, we describe the dynamics of the HIV reservoirs in vivo and how to detect them. We then introduce treatment strategies and new agents that have been reported to be effective in reducing or inactivating HIV reservoirs.

## 2. HIV Reservoirs in HIV-Infected Individuals Undergoing cART

After infection with HIV, proviral DNA is produced from HIV genomic RNA by the HIV reverse transcriptase, which is subsequently integrated into the host DNA. Certain HIV proviruses immediately start producing HIV RNA and proteins, whereas others remain silent as “HIV latently infected cells”. Latent HIV proviruses exist in all CD4^+^ T cell subsets; however, they exist predominantly in resting memory T cells [8,14,15,16]. As HIV reservoir cells do not produce HIV-related RNA and proteins, they are indistinguishable from uninfected cells and are not eliminated by immune cells. In 2017, CD32a was identified as a marker of latent HIV infection, but this was soon denied by another research group [17]. Therefore, detecting HIV reservoir cells using reservoir-specific markers is difficult. Roughly one in 10^5^–10^8^ CD4^+^ T cells are reported to be latently infected cells in patients with HIV undergoing cART, which indicates approximately 10^4^–10^7^ HIV latent cells in the whole body, assuming a total of 10^11^ CD4^+^ T lymphocytes [18,19]. Furthermore, only 1–2% of the lymphocytes in the body are believed to circulate in the blood. Most of the remaining lymphocytes are distributed in tissues throughout the body; however, their distribution varies [20]. Such conditions complicate the measurement of HIV reservoirs in the body. Lymphocytes in patients with HIV undergoing cART are mostly found in the lymph nodes, similar to those in uninfected people [21]. Thus, the presence of high concentrations of cells in the lymph nodes may increase the chances of the cell-to-cell transmission of HIV. In addition, the anatomical features of lymph nodes create pharmacological and immunological sanctuaries for HIV.

The reactivation level of cells latently infected with HIV via stimulation differs between blood and tissues, such as lymph nodes or the gastrointestinal tract. In addition, transcriptional activity in cells latently infected with HIV is affected by the provirus integration site [22,23,24,25,26,27,28]. Analyses of samples from elite controllers of HIV replication and patients under long-term cART revealed that full-length (replication-competent) proviruses were mainly found in transcriptionally inaccessible sites, such as centromeric satellite DNA and sites with heterochromatin features [28,29]. These results suggest that a less transcriptionally active reservoir is favored in both elite controllers and patients undergoing long-term ART. 

## 3. Detection and Measurement of HIV Reservoirs

In most cases, the only available sample from each patient was peripheral blood; however, as described previously, the detection and measurement of HIV latent reservoir cells from blood samples are challenging due to the limited number of reservoir cells. In addition, cell populations in the blood may not reflect the characteristics of reservoir cells throughout the body, particularly those in the lymph nodes, where the majority of lymphocytes are located. To detect cells latently infected with HIV in the blood, an easy and sensitive method involves the measurement of HIV proviral DNA in cells. Conventional quantitative polymerase chain reaction (qPCR) or droplet digital PCR can be used for quantification. DNA extracted from HIV-infected cells contains integrated HIV provirus and unintegrated provirus, which can be distinguished using different primer sets [30,31,32,33]. However, most HIV DNA from patients undergoing cART is considered to be in an integrated form [31]. Another issue in measuring HIV reservoir size using qPCR is that it detects not only intact (replication-competent) proviruses but also replication-incompetent (e.g., defective) proviruses [30,34]. This is because, in most cases, a primer set amplifies only a part of the HIV genome (such as the Gag region), and the PCR cannot distinguish between defective and full-length proviral genomes; thus, the HIV reservoir size quantified via DNA detection is likely much larger than the “true” reservoir size that can actually produce the virus. In our previous study using a cell-line-based model, we observed the accumulation of defective proviruses or replication-incompetent proviruses within a few months and found a cell clone with intact (without defection) proviruses that failed to produce HIV-1 mRNA in the model under treatment [35]. An analysis of clinical samples derived from HIV patients demonstrated that only 7% of proviruses expressed HIV mRNA from donors infected with HIV [36]. Furthermore, Anderson et al. showed that an increase in gag-deleted proviruses occurred only after 1–2 years of therapy, and the number of defective proviruses increased after 10–15 years of therapy [37].

Thus, another way to quantify HIV reservoirs is to stimulate peripheral lymphocytes ex vivo with an agent such as phorbol 12-myristate 13-acetate or phytohemagglutinin (PHA) and observe changes in the induction of HIV RNA transcription. This method allows for the detection of cells capable of inducing HIV RNA transcription. However, this method also detects proviruses that are not full length (replication competent) [30]. To detect only reservoir cells capable of producing HIV with replication-competent provirus, the quantitative viral outgrowth assay (QVOA) has been considered the gold-standard assay [9,26,31,38,39,40]. In this assay, CD4^+^ cells from a patient are activated ex vivo with agents such as PHA, anti-CD3, and anti-CD28 antibodies [41]. A positive result in this assay indicates the presence of HIV reservoir cells that can be reactivated and produce viruses; however, because the number of cells with replication-competent provirus in the blood is low in most patients undergoing cART, negative cases are often observed, even if they have some cells with the replication-competent provirus in the blood.

## 4. “Shock and Kill” Strategy and Drugs That May Eliminate HIV Reservoir Cells

The strategy to eliminate HIV reservoirs using latency-reversing agents (LRAs) has been referred to as “Shock and Kill” [42,43,44,45,46]. In this approach, cells latently infected with HIV are first reactivated with an LRA; reactivated cells start producing HIV-RNA, proteins, and HIV particles, and these cells are finally eliminated by immune effectors such as cytotoxic T lymphocytes or undergo virally induced cytopathic damage or apoptosis [35,47,48,49] (Figure 2). However, the process of reversing HIV latency using LRAs and eliminating HIV reservoir cells in vivo presents numerous obstacles. Several LRAs have been evaluated in clinical trials, with most studies demonstrating that the HIV latent reservoir size remains unchanged; thus, indicating that these drugs fail to reduce the number of cells latently infected with HIV in vivo [50,51,52,53,54,55]. These results imply that in vitro LRA potency does not necessarily correlate with clinical LRA potency, presumably because multiple mechanisms are involved in maintaining HIV latency in vivo [26,46,56,57]. A recent study showed that a significant proportion of T cell reservoir cells harboring replication-competent proviruses did not respond to stimulation-induced reactivation [26]. Moreover, hypothetically, the use of LRA alone in treatment-naïve patients with HIV could result in infectious virus production from reactivated latently infected cells, with subsequent infection of uninfected bystander cells. Therefore, combining LRA(s) with strong cART drugs is essential to effectively use LRA therapy to eliminate HIV reservoirs.

Several small-molecule drugs developed or reported to be LRAs (particularly in the early phase) include protein kinase C (PKC) activators (e.g., ingenol-3-angelate (PEP005), prostratin, and bryostatin-1), histone deacetylases (HDAC) inhibitors (e.g., SAHA/vorinostat), and BRD4 inhibitors (e.g., JQ1) [58,59]. The first LRA candidates to be clinically evaluated were HDAC inhibitors, including valproic acid and vorinostat (SAHA), some of which were originally developed as anticancer agents. Although numerous clinical studies investigating HDAC inhibitors observed a temporal increase in cell-associated and plasma HIV RNA following HIV latency reversal, they did not report a clear decrease in HIV reservoir cells [50,51,53,54,55,60]. Another concern with the use of HDAC inhibitors is that they (e.g., vorinostat) reportedly enhance CD4^+^ T cell susceptibility to HIV [61].

Another class of LRAs is agonists/activators of PKC, which play a critical role in the regulation of cell growth, differentiation, and apoptosis [62,63]. PKC agonists/activators induce transcription factors, such as NF-κB, which bind to the HIV long terminal repeat (LTR) and thus activate HIV mRNA transcription [64]. Recent studies have demonstrated that some PKC activators, such as PEP005, exhibit potent latency-reversing activity in several HIV-latent cell lines and primary cells derived from individuals infected with HIV in vitro [48,58,65,66,67]. A recent clinical trial using the PKC agonist bryostatin-1 demonstrated that the drug was safe in a single dose. However, the drug did not show any effect on latent HIV transcription in vivo, probably due to low plasma concentrations [52]. The potency of PKC activators, such as PEP005, is reported to be strongly enhanced by their combination with LRAs from a different class. Several groups have reported that combined treatment is effective for achieving maximum LRA reactivation [48,65,66], among which JQ1 combined with a PKC activator is to be considered one of the most effective combinations [48,49,65,66].

Antiviral cytotoxic T lymphocytes or antibodies are absent in conventional cell line-based models of cells latently infected with HIV in vitro (e.g., J-Lat). However, certain LRAs show latent cell-specific cytopathicity or apoptosis [35,68], and we have previously reported that some LRAs, such as PEP005, strongly induce the upregulation of active caspase-3, resulting in enhanced apoptosis [48,49]. HIV preferentially integrates into genomic regions via active transcription, resulting in a high proportion of HIV integration within the host genome [69]. In addition, the epigenetic circumstance of integrated proviruses is associated with the accessibility of transcription factors that drive the promoter activity of the 5ʹ-LTR. These findings indicate that LRA susceptibility among different latently infected with HIV varies depending on the genetic and epigenetic environments of the integrated proviruses. Thus, using such broad-acting LRAs alone (e.g., HDAC inhibitors or PKC activators) may be considered insufficient to eradicate HIV reservoirs in vivo, and the addition of other strategies/techniques with different mechanisms of action may be required. In this regard, Battivelli et al. [70] reported that LRAs could reactivate only some of the cells with latent proviruses using their in vitro model, thus demonstrating a wide variation in drug susceptibility to LRAs among different HIV-infected cells. Thus, drugs with different mechanisms of action or drug combinations that are effective in reproducing a wide variety of HIV proviruses must be developed to eliminate cells capable of producing HIV. Currently, three major models can provide preclinical testing for an HIV cure: in vitro primary cell or cell line systems, ex vivo testing in clinical samples, and in vivo animal models. The key determinant of the success of these models is whether they can recapitulate drug effects in vivo. In particular, genetic and epigenetic environments are key factors that determine the fate of HIV provirus, either active viral production or viral latency [24]. Currently available in vitro HIV latent models, such as ACH2, J1.1, and U1 cells, carry only one or two proviruses integrated into particular host genomic regions and cannot recapitulate the thousands of different integration sites observed in vivo [71,72]. Ex vivo models using clinical samples from HIV-infected individuals may also capitulate the diverse HIV integration sites [58]. However, the rarity of HIV-infected cells in the peripheral blood of HIV patients receiving cART [26,73] prevents a scalable and robust analysis of the effects of drugs. Although animal models provide valuable pre-clinical testing data, the associated costs and resources hinder large-scale screening [74,75,76]. Therefore, we recently developed a new in vitro infection model using Jurkat cells that harbor a much wider variety of HIV-infected cell clones than conventional in vitro models. This model also enables the analysis of both the antiviral effect of cART drugs and the effect of reservoir elimination by LRAs [35]. As another example of a primary cell-derived model, Dobrowolski et al. reported an HIV latency model (the QUECEL model), which can be used to successfully generate a large number of CD4^+^ T cells latently infected with HIV in vitro. A cocktail of cytokines including TGF-beta, IL-10, and IL-8 is used to generate the cell model with a homogeneous population of cells latently infected with HIV [77]. The LRA drug candidates identified via such cell line-based assay systems can be further evaluated via drug assays using primary CD4^+^ T cell-derived HIV latent reservoir models [77,78,79,80,81,82,83] or animal models (i.e., HIV-infected humanized mice or SIV-infected macaques) [74,75,76].

## 5. Strategy to Completely Inactivate HIV Reservoir Cells

Although the eradication of HIV involves the complete elimination of viral reservoirs, which has been an unattainable goal, further research towards a cure for HIV is underway. Developing a functional cure for HIV involves achieving persistent suppression of HIV transcription despite the presence of an integrated provirus via persistently silencing the latent provirus in infected cells. In this approach, proposed as a “Block and Lock” strategy (Figure 2), HIV reservoir cells with integrated HIV proviruses are not completely eradicated; however, viral transcription is kept at a low enough level that the produced virus can be cleared by the immune function. Multiple therapeutic targets have been investigated for inhibiting HIV transcription; however, they have not yet been examined in clinical trials due to the complexity of various factors involved in HIV transcription.

Currently, one of the most studied “Block and Lock” drugs that silences HIV transcription is the Tat inhibitor, diodehydro-cortistatin A (dCA) [84]. The Tat protein, which potently activates HIV gene expression [85], is highly conserved among HIV isolates, and has no cellular homologues. Tat is also an important factor in stimulating HIV transcriptional elongation via recruiting and activating RNA polymerase II. Mousseau et al. reported that dCA bound to the TAR-binding domain of Tat and inhibited transcriptional elongation of the HIV promoter, causing its epigenetic silencing [84]. Recent studies using CD4^+^ T cells from cART-treated individuals have shown that dCA induces a persistent latent infection that is resistant to reactivation by LRA, and that prior treatment with dCA slows and reduces viral rebound in a bone marrow/liver/thymus (BLT) mouse model [86]. Furthermore, when injected intraperitoneally into mice, dCA was shown to cross the blood–brain barrier and was detected at high levels in the brain, where microglial cells are thought to exist as HIV reservoirs [87]. These results suggest that dCA and other TAT inhibitors may be further developed for clinical use as they inhibit Tat-dependent transcription and induce a suppressive epigenetic circumstance that prevents HIV reactivation during treatment interruption.

Facilitating chromatin transcription complex (FACT) and bromodomain-containing protein 4 (BRD4) are important targets for HIV transcriptional regulators. FACT is an HIV transcriptional regulator consisting of a suppressor of Ty16 (SUPT16H) and a structure-specific recognition protein (SSRP1). FACT acts as a histone chaperone, destabilizing nucleosome structure and promoting RNA polymerase II-driven transcription. Previous studies have shown that the anticancer compound curaxin inhibits HIV replication and reactivation. Curaxin was also found to inhibit FACT and suppress NF-κB-mediated transcription, suggesting that curaxin might promote HIV latency via FACT inhibition [88]. Thus, a combination regimen with cART and a FACT inhibitor may lead to faster viremia reduction, a stronger suppression of HIV reactivation, and maintenance of deep HIV latency during treatment interruption. BRD4 is a member of the bromodomain and extra-terminal domain family and is involved in regulating the expression of various genes [89]. As mentioned previously, BRD4 inhibitors activate HIV transcription by promoting the binding of P-TEF-b and Tat, whereas BRD4 itself competitively inhibits their binding; thus, BRD4 regulators (enhancers) may be useful as HIV transcriptional repressors. ZL0580 is a recently developed BRD4 selective modulator that binds to BRD4 bromodomain 1 and inhibits HIV transcription [90]. The mechanism of action of ZL0580 is to prevent P-TEF-b (CDK9/cyclin T1 complex) binding to Tat by promoting BRD4-CDK9 binding, thereby preventing Tat binding to LTR [91]. Following treatment discontinuation, ZL0580 induces a delay in viral rebound (ex vivo) in the peripheral blood mononuclear cells (PBMCs) of aviremic individuals infected with HIV. Furthermore, it also inhibited spontaneous HIV replication in PBMCs from ART-naïve viremic individuals and inhibited PHA-stimulated reactivation [90]. Thus, the BRD4 modulator is a promising candidate as a new class of “Block and Lock” drugs; however, further validation is needed as BRD4 is a multifunctional molecule involved in HIV gene transcription.

The heat shock protein 90 (HSP90) is another potential therapeutic target. HSP90 is a cellular chaperone protein that facilitates the folding and stabilization of other proteins and protects cells, particularly under high temperature-induced stress [92]. HSP90 expression is known to increase in HIV-infected mononuclear cells and T cells, with increased protein production, including viral proteins [93,94]. Recent studies have shown that the HSP90 inhibitors, AUY922 and 17-AGG, prevent viral rebound in combination with the reverse transcriptase inhibitor EFdA after treatment interruption in HIV-infected humanized BLT mouse models. Viral reactivation was observed in PBMCs and the spleen upon restimulation, thus indicating that the HSP90 inhibitor was able to maintain the latency of the replication-competent HIV provirus [95]. These results suggest that HSP90 inhibitors may provide long-term remission of HIV replication owing to strong inhibition of HIV latent reservoirs.

Targeting cell signaling pathways, such as Janus Kinase (JAK) and mammalian target of rapamycin (mTOR), is also a possible option for “Block and Lock” therapy. The homeostasis of memory T cells, a sanctuary of the latent HIV reservoir, is regulated and maintained by cytokines, followed by JAK-STAT pathway activation [96]. Previous studies have reported that the JAK inhibitors ruxolitinib and tofacitinib inhibit HIV reactivation in primary CD4^+^ T cells, suggesting that the JAK/STAT pathway is involved in HIV persistence and reactivation [97]. These inhibitors, which have been approved for the treatment of blood disorders (myelofibrosis and polycythemia vera) and autoimmune diseases (rheumatoid arthritis, psoriatic arthritis, and ulcerative colitis), possess potent anti-inflammatory properties. In addition, ruxolitinib, which is known to exhibit anti-HIV activity in vitro, demonstrated efficacy and safety in a recent Phase 2 clinical trial (NCT02475655) in individuals infected with HIV on cART, including reductions in key markers associated with viral persistence [98]. The serine/threonine kinase mTOR forms two complexes, mTORC1 and m TORC2, which are involved in various cellular processes. The mTOR complex regulates HIV latency, and two mTORC1 inhibitor genes, TSC1 and DEPDC5, are involved in latent HIV infection [99,100]. Several clinical trials (NCT02990312 and NCT02440789) started in 2019 to evaluate the effects of the mTORC1 inhibitor sirolimus (rapamycin) on HIV latency, immune activation, and inflammation related to HIV infection.

In conclusion, these “Block and Lock” drug candidates show potential for development and can ultimately maintain HIV latent reservoirs or achieve a functional cure for HIV. However, further evaluation is required as most drugs directly affect molecules that may be associated with off-target effects via inhibiting or activating cellular factors which are critical for cell survival.

## 6. Non-Drug Strategies to Eliminate HIV Reservoir Cells

As another important curative strategy, anti-HIV broadly neutralizing antibodies (bNAbs) are being explored [101]. Antibodies are considered to be the key modulators of immunity and differ from cART in that they can recruit immune effector functions through their Fc domains [102]. Antibodies accelerate the clearance of viruses and infected cells, whereas antigen–antibody immune complexes are potent immunogens that can promote the development of host immune responses [102,103,104]. The potential of bNAbs to induce long-term remission of HIV has been shown using animal models [105,106]. Nishimura et al. evaluated the effects of the combination of 3BNC117 and 10–1074 in the absence of ART during early SHIV_AD8_ infection. In contrast to the ART condition, six out of thirteen animals achieved long-term viral control, and an additional four became elite controllers. Further, their analysis demonstrated that the antibody therapy facilitated the emergence of potent CD8^+^ T cell immunity that can durably suppress virus replication [105]. Borducchi et al. reported that the administration of an HIV env-specific broadly neutralizing antibody (PGT121) together with an LRA (Toll-like receptor 7 agonist: GS-9620) during cART successfully delayed viral rebound following the discontinuation of cART in simian human immunodeficiency virus (SHIV)-infected rhesus monkeys [106]. In humans, limited data are available on the effects of bNAbs on the latent HIV reservoirs. Some studies failed to show an effect of bNAbs on the HIV reservoir size in the treated patients [107,108]. However, another study showed that the combination of 3BNC117 and 10–1074 led to longer periods of viral suppression after ART discontinuation, including two individuals that continued to maintain suppression for over 30 weeks [109,110].

Other strategies, such as gene therapy using Clustered Regularly Interspaced Short Palindromic Repeats (CRISPR), immunotherapy, and hematopoietic stem cell transplantation (HSCT), have also been reported [111,112,113,114]. For immunotherapy, Liu et al. reported a clinical study for bNAb-derived chimeric antigen receptor (CAR) T cells, with a total of fourteen participants who received a single administration of bNAb-derived CAR-T cells. In this study, six participants discontinued ART, and viremia rebound occurred in all of them, with a 5.3-week median time. However, cell-associated viral RNA and intact proviruses decreased significantly after CAR-T cell treatment; thus, suggesting that CAR-T cells exerted pressure on rebound viruses [111]. As some patients with cancer who received HSCT maintained long-term HIV remission, HSCT has been considered a potential treatment option to cure HIV. However, the detailed mechanism and optimal protocol remain to be developed. Lastly, Wu et al. showed using SIV-infected macaque model that the allogeneic immunity cleared HIV latent cells following stem cell transplantation [112].

## 7. Perspectives towards the HIV Cure

Over the last 20 years, numerous anti-HIV drugs targeting different stages of the HIV life cycle have become available. The development of effective antiviral agents has improved our ability to manage HIV infections. Furthermore, the introduction of cART with a single-tablet regimen and long-acting drugs has significantly changed the quality of life of patients with HIV. However, as discussed in this review, continuous efforts have been made to establish a way to eliminate HIV reservoirs and consolidate latency conditions toward the cure of HIV infection, which is one of the most important and challenging topics in current HIV research (Figure 1).

After reactivation using the “Shock and Kill” strategy, an adequate immune response is required for the effective elimination of HIV reservoir cells. To effectively “kill,” some LRAs have been reported to induce apoptosis, specifically in cells latently infected with HIV after reactivation [35,68]. Furthermore, some drugs, such as second mitochondrial-derived activator of caspases (SMAC), mimetics, and toll-like receptor agonists, are expected to exhibit LRA activity and also activate the immune system (e.g., via inhibiting cell survival mechanisms) [47,68].

Unlike the “Shock and Kill” strategy, the “Block and Lock” strategy attempts to permanently silence the HIV provirus in reservoir cells and strengthen the HIV latency state. This strategy is expected to prevent or delay viral rebound after cART interruption [91]. For a cure using both of these strategies, further studies using in vivo HIV latent infection models and clinical trials are needed. These strategies have completely different actions on HIV reservoir cells and they may be effective in ultimately developing a functional cure for HIV, if used sequentially to target distinct subsets of HIV reservoir cells: initial reactivation of LRA-sensitive cell populations is expected to reduce reservoir size via “Shock and Kill.” Subsequently, “Block and Lock” is applied to reservoir cells which are non-responsive to LRA; thus, causing reservoir cells to fall into a deep latency that cannot be easily reactivated [35,86,99].

Although permanently silencing all proviruses with a “Block and Lock” strategy is difficult, extending the period of viral rebound may be possible; however, determining the duration of rebound suppression using this strategy requires further investigation. Furthermore, the side effects of long-term administration of “Block and Lock” drugs and their effects on the immune system represent the main issues that require long-term analysis in vitro and subsequent clinical trials. Numerous patients with HIV and their doctors will consider it to be dangerous to discontinue cART after “Block and Lock” treatment; thus, continuing both therapies is an option. The benefit of this is that chronic inflammation seen in cART-treated patients may improve by completely suppressing the production of virus or viral proteins during cART treatment.

Needless to say, not only these two strategies, but also other treatment options, such as those discussed in Section 6, are very important. For example, during the reservoir removal phase in “Shock and Kill”, the removal of reservoir cells is likely to be more effective if bNAbs and/or immunomodulators are administered simultaneously. Therapies using CRISPR or CAR-T may not be fully effective on their own, but when combined with other therapies, they are likely to be more effective in the removal and inactivation of HIV reservoirs. However, we will need to wait for the results of future clinical studies to know what combinations in what order are most effective. Furthermore, the quality and quantity of the residual reservoirs may be widely different among patients, and the development of methods to accurately identify these differences is critical to controlling HIV reservoirs in each patient.

The progress in HIV research since the 1980s has been remarkable. However, we have not yet reached a cure. As knowledge and technology advance, new findings will eventually lead to breakthroughs, and more detailed future research on the HIV reservoirs will provide valuable insights into future treatment strategies that might cure HIV.

## Figures and Tables

**Figure 1 ijms-25-02621-f001:**
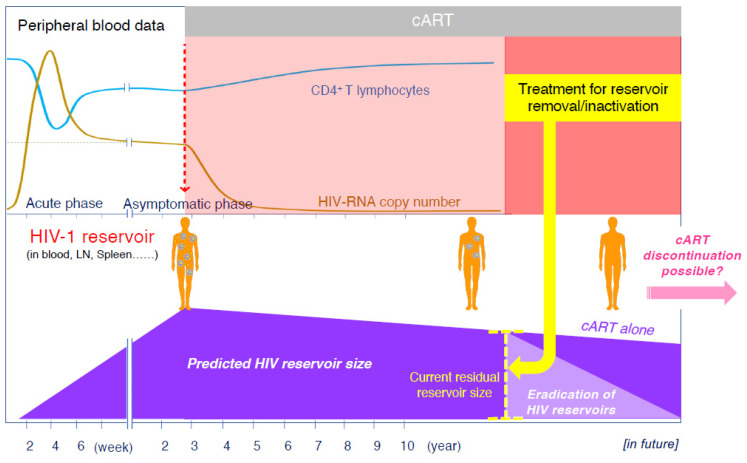
Clinical course of a patient with HIV receiving cART and predicted changes in the residual HIV reservoir size in the body of the patient. Yellow dots in the human body shown in the figure represent the virus and blue circles represent HIV-infected cells.

**Figure 2 ijms-25-02621-f002:**
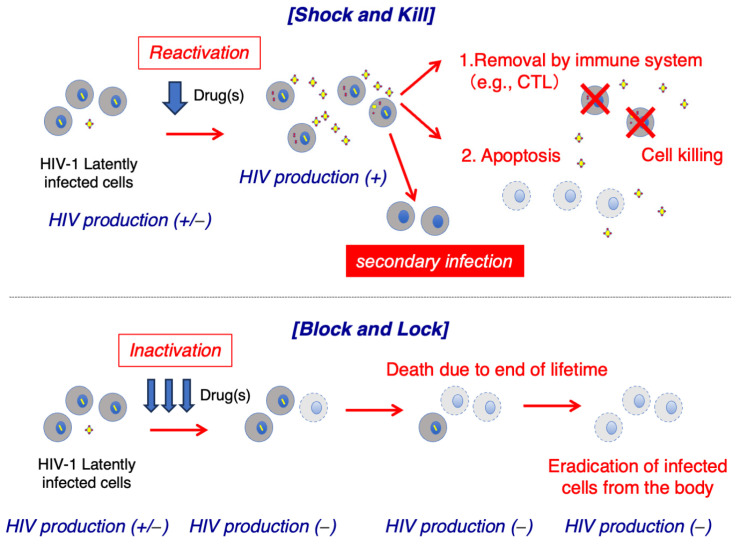
Scheme of the “Shock and Kill” and “Block and Lock” strategies.

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
