# Peer review of "HIV Reservoirs and Treatment Strategies toward Curing HIV Infection"

_ijms, 2024, doi:10.3390/ijms25052621_

Round 1

Reviewer 1 Report

Comments and Suggestions for Authors

The review article “Current status of therapeutic development for treating viral reservoirs toward curing HIV infections” by Matsuda and Maeda aims to provide an overview on the two stratgiees “Shock & Kill”, and “Block & Lock” that are being pursued to eliminate the integrated provirus thus reducing the reservoir size and helping in HIV control. In my opinion the articles falls short of compiling the data to provide the readers with enriched information. There are several articles that highlight the various approaches in combination of different broadly neutralizing antibodies that are not discussed. Additionally, approaches targeting the host proteins to purge the infected cells have been published but not mentioned in this article. The “block and kill” lacks discussion from the author’s perspective regarding the long-term impact of using the inhibitors/enhancers that target host genes. Overall, the article lacks presentation and interest.

Author Response

Reviewer 1:

The review article “Current status of therapeutic development for treating viral reservoirs toward curing HIV infections” by Matsuda and Maeda aims to provide an overview on the two stratgiees “Shock & Kill”, and “Block & Lock” that are being pursued to eliminate the integrated provirus thus reducing the reservoir size and helping in HIV control. In my opinion the articles falls short of compiling the data to provide the readers with enriched information. There are several articles that highlight the various approaches in combination of different broadly neutralizing antibodies that are not discussed.

Reply: Thank you for your constructive suggestions, I agree that anti-HIV bNAbs are important to reduce HIV reservoir cells, and the original manuscript (page 5, line 169~) also mentioned studies investigating the effects of PGT121 and 3BNC117. In the revised manuscript, we have modified that paragraph and created a new section addressing the use of antibodies and immunotherapy in HIV treatment. Furthermore, we discussed the effects of combinations of multiple antibodies in clinical trials.

(Page 5) 5. Non-drug strategies to eliminate HIV reservoir cells

As another important curative strategy, anti-HIV-broadly neutralizing antibodies (bNAbs) are being explored70. Antibodies are considered to be the key modulators of immunity and differ from cART in that they can recruit immune effector functions through their Fc domains71. Antibodies accelerate the clearance of viruses and infected cells, whereas antigen-antibody immune complexes are potent immunogens that can promote the development of host immune responses71-73. The potential of bNAbs to induce long-term remission of HIV has been shown using animal models 74, 75. Nishimura et al. evaluated the effects of the combination of 3BNC117 and 10-1074 in the absence of ART during early SHIVAD8 infection. In contrast to the ART condition, 6 out of 13 animals achieved long-term viral control, and an additional 4 became elite controllers. Further, their analysis demonstrated that the antibody therapy facilitated the emergence of potent CD8+ T cell immunity that can durably suppress virus replication74. Borducchi et al. reported that the administration of an HIV env-specific broadly neutralizing antibody (PGT121) together with an LRA (Toll-like receptor 7 agonist: GS-9620) during cART successfully delayed viral rebound following the discontinuation of cART in simian human immunodeficiency virus (SHIV)-infected rhesus monkeys75. In humans, limited data are available on the effects of bNAbs on the latent HIV reservoirs. Some studies failed to show the effect of bNAbs on the HIV reservoir size in the treated patients76, 77. However, another study showed that the combination of 3BNC117 and 10-1074 led to longer periods of viral suppression after ART discontinuation, including two individuals that continued to maintain suppression for over 30 weeks78, 80.

Other strategies, such as gene therapy using Clustered Regularly Interspaced Short Palindromic Repeats, immunotherapy, and hematopoietic stem cell transplantation (HSCT), have also been reported79, 81-83. For immunotherapy, Liu et al. reported a clinical study for bNAb-derived chimeric antigen receptor (CAR) T cells, with a total of 14 participants who received a single administration of bNAb-derived CAR-T cells. In this study, six participants discontinued ART, and viremia rebound occurred in all of them, with a 5.3-week median time. However, cell-associated viral RNA and intact proviruses decreased significantly after CAR-T cell treatment; thus, suggesting that CAR-T cells exerted pressure on rebound viruses79. As some patients with cancer who received HSCT maintained long-term HIV remission, HSCT has been considered a potential treatment option to cure HIV. However, the detailed mechanism and optimal protocol remain to be developed. Lastly, Wu et al. showed using SIV-infected macaque model that the allogeneic immunity cleared HIV latent cells following stem cell transplantation81.

.

Additionally, approaches targeting the host proteins to purge the infected cells have been published but not mentioned in this article. The “block and kill” lacks discussion from the author’s perspective regarding the long-term impact of using the inhibitors/enhancers that target host genes. Overall, the article lacks presentation and interest.

Reply: Thank you for your suggestion. This review mainly concentrates on pharmacotherapy strategies for targeting the HIV reservoir cells; thus, making it challenging to include discussions on all HIV-related articles. However, as you point out, there are host proteins, antibodies, immunotherapy, gene therapy, and various other important approaches known to contribute to HIV reservoir reduction, some of which are briefly discussed in this review.

As per your suggestion, the side effects of long-term administration of Block & Lock drugs are a crucial issue. We added the following text to the revised manuscript.

(Page 9, Line 362) Although permanently silencing all proviruses with a “Block & Lock” strategy is difficult, extending the time for viral rebound may be possible; however, determining the time duration of rebound suppression using this strategy requires further investigation. Furthermore, the side effects of long-term administration of “Block & Lock” drugs and their effects on the immune system represent the main issues that require long-term analysis in vitro and subsequent clinical trials.

Reviewer 2 Report

Comments and Suggestions for Authors

The research question is important. As the authors themselves wrote in the Abstract, combination antiretroviral therapy (cART) has significantly improved the prognosis of individuals living with human immunodeficiency virus (HIV) and acquired immunodeficiency syndrome has gone from a fatal disease to a treatable chronic infection. The manuscript is a review focusing on drug development targeting latently HIV-infected cells and an overview of progress, reviewing articles in relation to the “Shock & Kill” and “Block & Lock” strategies. In this sense, the review is relatively well written, addresses relevant articles on the topic and presents interesting illustrations, enabling a good understanding of the state of the art on this specific topic.

However, I have two main concerns:

(1) There are several similar reviews recently published on this subject, including some in the MDPI scientific journals. So it's not original.

(2) The manuscript focuses on just some of the treatment alternatives for curing HIV. A more complete description of the general scenario of possible therapies is lacking. In other words, I consider that it could be broader/complete, including gene therapy, immunotherapy and hematopoietic stem cell transplantation (HSCT) procedures. I would suggest a specific section to other therapies or they could be described better in the perspectives section. Finally, this last section (perspectives) should be really improved to include a more general view of the HIV treatment and not only repeating “Shock & Kill” and “Block & Lock” .

Comments on the Quality of English Language

 Minor editing of English language required

Author Response

Reviewer 2:

The research question is important. As the authors themselves wrote in the Abstract, combination antiretroviral therapy (cART) has significantly improved the prognosis of individuals living with human immunodeficiency virus (HIV) and acquired immunodeficiency syndrome has gone from a fatal disease to a treatable chronic infection. The manuscript is a review focusing on drug development targeting latently HIV-infected cells and an overview of progress, reviewing articles in relation to the “Shock & Kill” and “Block & Lock” strategies. In this sense, the review is relatively well written, addresses relevant articles on the topic and presents interesting illustrations, enabling a good understanding of the state of the art on this specific topic.

Reply: Thank you for your comment. Although there are several review articles on HIV reservoirs, we believe this paper will be of interest to a large number of readers.

However, I have two main concerns:

(1) There are several similar reviews recently published on this subject, including some in the MDPI scientific journals. So it's not original.

Reply: Thanks for pointing that out, there have been several reviews published on the HIV reservoir cells and potential treatments. However, I believe most of them are specific to one treatment approach such as antibody therapy, shock & kill, block & lock, etc. This review focuses on drug-based therapies, and, as mentioned in the discussion, it also discusses the combination of Shock & Kill followed by Block & Lock, which I believe will provide new insights into future treatment strategies for HIV reservoirs.

(Page 8, Line 355) These strategies have completely different actions on HIV reservoir cells and they may be effective in ultimately developing a functional cure for HIV, if used sequentially to target distinct subsets of HIV reservoir cells: initial reactivation of LRA-sensitive cell populations is expected to reduce reservoir size via “Shock & Kill.” Subsequently, “Block & Lock” is applied to reservoir cells irresponsive to LRA, thus causing reservoir cells to fall into deep latency that cannot be easily reactivated.

(2) The manuscript focuses on just some of the treatment alternatives for curing HIV. A more complete description of the general scenario of possible therapies is lacking. In other words, I consider that it could be broader/complete, including gene therapy, immunotherapy and hematopoietic stem cell transplantation (HSCT) procedures. I would suggest a specific section to other therapies or they could be described better in the perspectives section.

Reply: We thank for your very constructive suggestions. Although this review article focuses primarily on drugs, I think it would be worthwhile to cover antibodies, immunotherapy, gene therapy, and stem cell transplantation. We have therefore created a new section to describe these.

(Page 5) 5. Non-drug strategies to eliminate HIV reservoir cells

As another important curative strategy, anti-HIV-broadly neutralizing antibodies (bNAbs) are being explored70. Antibodies are considered to be the key modulators of immunity and differ from cART in that they can recruit immune effector functions through their Fc domains71. Antibodies accelerate the clearance of viruses and infected cells, whereas antigen-antibody immune complexes are potent immunogens that can promote the development of host immune responses71-73. The potential of bNAbs to induce long-term remission of HIV has been shown using animal models 74, 75. Nishimura et al. evaluated the effects of the combination of 3BNC117 and 10-1074 in the absence of ART during early SHIVAD8 infection. In contrast to the ART condition, 6 out of 13 animals achieved long-term viral control, and an additional 4 became elite controllers. Further, their analysis demonstrated that the antibody therapy facilitated the emergence of potent CD8+ T cell immunity that can durably suppress virus replication74. Borducchi et al. reported that the administration of an HIV env-specific broadly neutralizing antibody (PGT121) together with an LRA (Toll-like receptor 7 agonist: GS-9620) during cART successfully delayed viral rebound following the discontinuation of cART in simian human immunodeficiency virus (SHIV)-infected rhesus monkeys75. In humans, limited data are available on the effects of bNAbs on the latent HIV reservoirs. Some studies failed to show the effect of bNAbs on the HIV reservoir size in the treated patients76, 77. However, another study showed that the combination of 3BNC117 and 10-1074 led to longer periods of viral suppression after ART discontinuation, including two individuals that continued to maintain suppression for over 30 weeks78, 80.

Other strategies, such as gene therapy using Clustered Regularly Interspaced Short Palindromic Repeats, immunotherapy, and hematopoietic stem cell transplantation (HSCT), have also been reported79, 81-83. For immunotherapy, Liu et al. reported a clinical study for bNAb-derived chimeric antigen receptor (CAR) T cells, with a total of 14 participants who received a single administration of bNAb-derived CAR-T cells. In this study, six participants discontinued ART, and viremia rebound occurred in all of them, with a 5.3-week median time. However, cell-associated viral RNA and intact proviruses decreased significantly after CAR-T cell treatment; thus, suggesting that CAR-T cells exerted pressure on rebound viruses79. As some patients with cancer who received HSCT maintained long-term HIV remission, HSCT has been considered a potential treatment option to cure HIV. However, the detailed mechanism and optimal protocol remain to be developed. Lastly, Wu et al. showed using SIV-infected macaque model that the allogeneic immunity cleared HIV latent cells following stem cell transplantation81.

Finally, this last section (perspectives) should be really improved to include a more general view of the HIV treatment and not only repeating “Shock & Kill” and “Block & Lock” .

Reply: Thank you for pointing this out. As per your suggestions, we revised this section and provided an overview picture.

(Page 9, Line 372) The progress in HIV research since the 1980s has been remarkable. However, we have not yet reached a cure. As knowledge and technology advance, new findings will eventually lead to breakthroughs, and more detailed future research on the HIV reservoir will provide valuable insights into future treatment strategies that might cure HIV.

Round 2

Reviewer 2 Report

Comments and Suggestions for Authors

As I have already stated before, the research question is important. The review is relatively well written, addresses relevant articles on the topic and presents interesting illustrations, enabling a good understanding of the state of the art on this specific topic. I had also manifested two main concerns in my first review:

(1) There are several similar reviews recently published on this subject, including some in the MDPI scientific journals. So it's not original.

(2) The manuscript focuses on just some of the treatment alternatives for curing HIV. A more complete description of the general scenario of possible therapies is lacking. In other words, I consider that it could be broader / more complete, including gene therapy, immunotherapy and hematopoietic stem cell transplantation (HSCT) procedures. I suggested a specific section to other therapies. Finally, this last section (perspectives) should be really improved to include a more general view of the HIV treatment and not only to repeat “Shock & Kill” and “Block & Lock”.

In this new version (R1), the authors added a new section (5. Non-drug strategies to eliminate HIV reservoir cells ) to meet my second recommendation. So the review was improved, although the main subject has already been reviewed by different previously published articles.

Excluding this issue of originality, I think the article still needs some additional adjustments, as described below:

1)   Please rewrite the objectives of the manuscript in the Introduction (lines 38 to 40: In this article, we focus on developing therapeutic agents with novel mechanisms of action that target cells latently infected with HIV as the ultimate goal for treatment of HIV infections). It is important to clarify the main objectives of the review article in the last sentence of the Introduction. Remember: you didn't develop anything and it is necessary to cite the different sections of the review (including the viral dynamics in the host and HIV-1 detection, since both subjects are in sections 2 and 3).

2)   I would suggest to rename the titles. Please avoid the words “Development” in section 5 and “Discussion” in section 7). I would recommend to think in more generic titles to include the main approached subjects in each section.

3)   I also strongly suggest to reorder the sections, since the authors are presenting “strategies with drugs” in the sections 4 and 6. On contrary, they are describing “non-drugs strategies” on section 5. I think it would be much more logical to order the sections as follows: 4, 6 and 5.

4)   Please revise carefully the section 7. I think many of the subjects could be relocated to other sections. And this last section should be short and direct to define “perspectives toward HIV-1 cure". 

Comments on the Quality of English Language

Moderate editing of English language required

Author Response

As I have already stated before, the research question is important. The review is relatively well written, addresses relevant articles on the topic and presents interesting illustrations, enabling a good understanding of the state of the art on this specific topic. I had also manifested two main concerns in my first review:

(1) There are several similar reviews recently published on this subject, including some in the MDPI scientific journals. So it's not original.

(2) The manuscript focuses on just some of the treatment alternatives for curing HIV. A more complete description of the general scenario of possible therapies is lacking. In other words, I consider that it could be broader / more complete, including gene therapy, immunotherapy and hematopoietic stem cell transplantation (HSCT) procedures. I suggested a specific section to other therapies. Finally, this last section (perspectives) should be really improved to include a more general view of the HIV treatment and not only to repeat “Shock & Kill” and “Block & Lock”.

In this new version (R1), the authors added a new section (5. Non-drug strategies to eliminate HIV reservoir cells ) to meet my second recommendation. So the review was improved, although the main subject has already been reviewed by different previously published articles.

(Response) Thank you for pointing this out. We referred several review articles on reservoir and drug development (they are cited). We have selected important drugs and obtained information from multiple review and original articles, and described them to fit the style of our manuscript. Thus, we are confident that the overall originality is maintained and this paper is not a copy of any particular previous article. In addition, in the revised manuscript, we have changed the wording throughout the text (please see “Matsuda_ijms-revision_R2_w_correction” file).

Excluding this issue of originality, I think the article still needs some additional adjustments, as described below:

1)   Please rewrite the objectives of the manuscript in the Introduction (lines 38 to 40: In this article, we focus on developing therapeutic agents with novel mechanisms of action that target cells latently infected with HIV as the ultimate goal for treatment of HIV infections). It is important to clarify the main objectives of the review article in the last sentence of the Introduction. Remember: you didn't develop anything and it is necessary to cite the different sections of the review (including the viral dynamics in the host and HIV-1 detection, since both subjects are in sections 2 and 3).

(Response) Thanks for the suggestion. I think you are correct. We have rewritten that part as follows:

“In this article, we describe the dynamics of the HIV reservoir in vivo and how to detect it. We then introduce treatment strategies and new agents that have been reported to be effective to reduce or inactivate HIV reservoirs."

2)   I would suggest to rename the titles. Please avoid the words “Development” in section 5 and “Discussion” in section 7). I would recommend to think in more generic titles to include the main approached subjects in each section.

(Response) Thank you very much. I came up with the following title:

(Title) HIV reservoirs and treatment strategies toward curing HIV infection

Section titles were also changed as indicated:

(Section 4) “Shock and Kill” strategy and drugs that may eliminate HIV reservoir cells.

(Section 7) Perspectives toward HIV cure.

If you have any suggestions, I'd appreciate it if you let me know.

3)   I also strongly suggest to reorder the sections, since the authors are presenting “strategies with drugs” in the sections 4 and 6. On contrary, they are describing “non-drugs strategies” on section 5. I think it would be much more logical to order the sections as follows: 4, 6 and 5.

 (Response) Thanks you. These sections were reordered according to your suggestion.

4)   Please revise carefully the section 7. I think many of the subjects could be relocated to other sections. And this last section should be short and direct to define “perspectives toward HIV-1 cure". 

 (Response) Thank you for your suggestions, we shortened the section 7 by moving a long sentence on the LRA assays to section 4 (Shock & Kill section). Also, we have removed multiple sentences that were repeated in other sections or inside section 7 (see “Matsuda_ijms-revision_R2_w_correction” file).

Finally, section title was changed as indicated:

(Section 7) Perspectives toward HIV cure.

Thank you again for your kind and constructive suggestions.
